# Microscopic Views of Atomic and Molecular Oxygen Bonding with *epi* Ge(001)-2 × 1 Studied by High-Resolution Synchrotron Radiation Photoemission

**DOI:** 10.3390/nano9040554

**Published:** 2019-04-04

**Authors:** Yi-Ting Cheng, Hsien-Wen Wan, Chiu-Ping Cheng, Jueinai Kwo, Minghwei Hong, Tun-Wen Pi

**Affiliations:** 1Graduate Institute of Applied Physics and Department of Physics, National Taiwan University, Taipei 10617, Taiwan; ceo6120@gmail.com (Y.-T.C.); b00202022@ntu.edu.tw (H.-W.W.); 2Department of Electrophysics, National Chiayi University, Chiayi 60004, Taiwan; 3Department of Physics, National Tsing Hua University, Hsinchu 30013, Taiwan; 4National Synchrotron Radiation Research Center, Hsinchu 30076, Taiwan

**Keywords:** Ge(001)-2 × 1, oxidation, synchrotron radiation photoemission

## Abstract

In this paper, we investigate the embryonic stage of oxidation of an *epi* Ge(001)-2 × 1 by atomic oxygen and molecular O_2_ via synchrotron radiation photoemission. The topmost buckled surface with the up- and down-dimer atoms, and the first subsurface layer behaves distinctly from the bulk by exhibiting surface core-level shifts in the Ge 3d core-level spectrum. The O_2_ molecules become dissociated upon reaching the *epi* Ge(001)-2 × 1 surface. One of the O atoms removes the up-dimer atom and the other bonds with the underneath Ge atom in the subsurface layer. Atomic oxygen preferentially adsorbed on the *epi* Ge(001)-2 ×1 in between the up-dimer atoms and the underneath subsurface atoms, without affecting the down-dimer atoms. The electronic environment of the O-affiliated Ge up-dimer atoms becomes similar to that of the down-dimer atoms. They both exhibit an enrichment in charge, where the subsurface of the Ge layer is maintained in a charge-deficient state. The dipole moment that was originally generated in the buckled reconstruction no longer exists, thereby resulting in a decrease in the ionization potential. The down-dimer Ge atoms and the back-bonded subsurface atoms remain inert to atomic O and molecular O_2_, which might account for the low reliability in the Ge-related metal-oxide-semiconductor (MOS) devices.

## 1. Introduction

Due to their high carrier mobilities, both the Ge and III-V compound semiconductors are channel materials that might replace silicon in p- and n-type metal-oxide-semiconductor field-effect transistors (MOSFETs) [1,2,3,4,5,6,7,8,9,10]. For the III-V metal-oxide-semiconductor (MOS), the established reports clearly show that a high-quality oxide/(In)GaAs interface leads to high-performance (In)GaAs MOSFETs in the drain currents and transconductances [9,10,11,12,13]. The premise of a high-quality interface requires a high-quality III-V surface (e.g., impurity-free and with a long-range order reflecting the nominal surface-atomic structure). Namely, the researchers are obligated to grow a high-κ dielectric oxide onto a well-defined reconstructed III-V surface. Years of researches on (In)GaAs(001) have concluded that chemical processes to the hetero-element surfaces would destroy the long-range order; the removal of the native oxides is out of the question for attaining a high-quality high-κ/III-V interface. To be specific, the as-grown (In)GaAs surface from a molecular-beam epitaxy (MBE) chamber or a metal organic chemical vapor deposition (MOCVD) reactor must be placed in direct contact with a dielectric oxide that was prepared with MBE or atomic layer deposition (ALD) without exposing it to air. For Ge, the surface pretreatment with assorted chemicals is typically done before dielectric deposition. The confidence in pretreatment is fully derived from the success of Si, where surface treatments are necessary in further device fabrication. Sun et al. have demonstrated that the Ge(001) surface becomes noticeably rough after exposure to aqueous HF and HCl solutions [14]. This raises concerns regarding the failure to advance Ge MOSFETs, which is presumably due to developers overlooking surface quality issues. Moreover, the engineering and surface-science communities, especially the synchrotron-radiation circle, have independently fought their wars, failing to maintain close communications with each other, thereby hindering efforts to achieve sufficient understanding of the interfaces.

The study of GeO_2_/Ge becomes inevitable as a result of the desire to obtain an interface that is similar to that of SiO_2_/Si with the oxide downsized to a few atomic layers. Researchers on Ge have been naturally inclined to take the established Si knowledge to propose the Ge MOS structure, and intensive research [15,16,17,18,19,20] and industry-scale development efforts in the Ge MOS have been undertaken to realize the product. Nevertheless, there is a dearth of fundamental research on Ge, largely because the Ge(001)-2 × 1 surface undergoes a similar reconstruction to that of Si(001)-2 × 1 and the surface and interfacial behaviors of Ge(001)-2 × 1 are believed to be no different from those of Si(001)-2 × 1. To our knowledge, this belief has never been questioned. In fact, the surface electronic structure of Ge(001)-2 × 1 reveals undeniable differences from that of Si(001)-2 × 1, where synchrotron radiation photoelectron spectroscopy (SRPES) shows the latter to exhibit distinct surface behaviors down to the third layer [21,22], but the former only to the second layer at room temperature (Figure 1) [23].

The lack of a detailed understanding regarding the Ge(001)-2 × 1 surface is an obstacle to understanding why Ge(001)-2 × 1 is unlikely to produce 1+ to 4+ charge states, as Si(001)-2 × 1 does [24]. The reported four Ge charge states were commonly observable by neutral beam oxidation [25], through chemically wetted treatments [26], during acid processes [27,28], or upon sudden exposure to a large amount of atomic oxygen on a 2 × 1 reconstructed surface [29]. It is odd to see that these thick Ge oxides respond differently with heat, with one showing an increase [30] and the other showing a decrease [27] in the strength of GeO_2_ with increasing annealing temperatures. Nevertheless, the investigation of a thick Ge oxide film does not help us to fully understand the oxygen-contacted Ge-Ge dimers layer. Another method, an in vacuo process using supersonic molecular oxygen beams, only produces the 1+ and 2+ oxidation states in the sub-monolayer thickness [31]. The weak bonding of molecules on Ge(001) [32,33] and the confined adsorption site at the surface dimers without any insertion into the Ge backbonds [34] have rendered the four charge states unlikely to be formed at the interfacial region of Ge(001).

O_2_ is introduced into the MBE chamber to compensate for the loss of oxygen from bombarding the oxide target upon the deposition of a high-κ dielectric oxide onto a Ge(001) wafer,. The partial pressure of O_2_ is unreactive to the (In)GaAs substrates [35], but it is not entirely unaffected at the Ge substrate; the ALD is operating at a high pressure and it generates similar concerns for the oxygen residual. The established records of initial O_2_ adsorption on Ge(001) have made their comments based on the very first work by Fukuda and Ogina two decades ago, who suggested that one of the dissociated O atoms of O_2_ sits at the bridge site and the other sits at a backbond of the Ge-Ge dimer [34,36]. However, the nondissociative O_2_ adsorption has also been experimentally and theoretically proposed later in the literature [33,37,38]. Nevertheless, the established records fall short of experimental investigations on the interfacial electronic structure of Ge(001) with atomic O and molecular O_2_, especially at the embryonic stage of adsorption. Worse yet, the investigations regarding an *epi* Ge(001)-2 × 1 surface are practically non-existent in the literature, the lack of which prevents us from improving the growth processes.

In this work, the *epi* Ge(001)-2 × 1 samples were exposed to atomic oxygen and high-purity O_2_ at dosages from as small as 0.06 Langmuir (L) to as high as 400 L. SRPES probed the O/Ge system, which has been identified as a superb tool to probe the atom-to-atom interactions at the interface. The tunability of photon energies allows us to acquire the spectra with high surface sensitivity, which cannot be realized by conventional X-ray spectroscopy (XPS). Note that the present work does not deny the established XPS works, because, at its resolution, it is not possible to trace a surface component with a surface core-level shift (SCLS) as small as 150 meV. At a low concentration, the SRPES can provide the response of each atom in a buckled dimer to oxygen. For O_2_/Ge(001), we found that the O_2_ molecules become dissociated upon reaching the Ge surface. One of them would remove the up-dimer atom from the surface and the other bonds with the Ge atom in the subsurface layer. Two oxygen-induced chemical states are then resolved in the Ge 3d and O 1s core-level spectra. For atomic oxygen on Ge(001), only the charge-enriched up-dimer atoms will accept O into the vicinity, whereas the charge-deficient down-dimer atoms remain inert to O. Furthermore, the up-dimer atom donates its excess charge to the bonded oxygen atom under the intact dimer bond. Consequently, the charge environment becomes similar to that of the down-dimer atom. The present experimental results provide direct evidence that the unpassivated down-dimer atoms might account for the reliability issue that is related to Ge MOS devices [39,40].

## 2. Materials and Methods

### 2.1. Sample Preparations

The MBE technique was used to grow *epi* Ge(001)-2 × 1 layers on a Ge substrate. The Sb-doped n-type Ge(001) wafer with a resistivity of 0.31–0.34 Ω-cm was dipped in 2% diluted HF solution and then rinsed in de-ionized water. The wafer was then immediately loaded into a multi-chamber MBE/analysis system [41], in which it was annealed in a UHV at 600°C for 20 min to achieve an ordered surface. The quality of the sample was assessed based upon the streaky reconstructed reflection high-energy electron diffraction (RHEED) patterns [42]. Afterwards, 7-nm thick Ge was grown on the annealed Ge substrate while using an effusion cell in an MBE chamber at a UHV pressure of less than of 2 × 10^−10^ Torr. Using this approach, the Ge epi-layer shows a greatly improved 2× RHEED pattern with streaky diffraction spots and more distinct Kikuchi arcs. The sharper and more intense diffraction patterns indicate the attainment of a morphologically flatter and more atomically ordered surface [2]. The samples were stored in vacuo in a portable UHV module to transport to the nearby National Synchrotron Radiation Research Center for SRPES measurements. The photoelectrons were collected with a 150-mm hemispherical analyzer (SPECS, GmbH) in a µ-metal chamber with a base pressure that is better than 1.2 × 10^−10^ Torr. The overall instrumental resolution was greater than 60 meV. Silver film freshly grown from an electron gun before the measurements was used to determine the energy reference. Atomic oxygen was generated through a commercial cracker (SPECS, GmbH), and length of time on cracking and the chamber pressure determined the dosage (L).

### 2.2. Data Analysis

The objective of SRPES experiments on semiconductor surfaces is to relate the observable features with the known properties of the reconstructed surfaces. A Voigt function line commonly represents the photoemission component; that is, it is a convolute of the Lorentzian and Gaussian functions. A model function that is assumed to consist of a multiplicity of overlapping components should be set up and fit accordingly to the data by the least squares method. Constraints are necessary to reduce the ambiguity of a fit, such as the lifetime width, spin-orbit splitting, and ratio essentially identical for all of the components. The three parameters are, for each spin-orbit pair, its position, height, and Gaussian width. The amplitude of the line is, in principle, related to the areal density and the location of the atoms within the surface layer through the inelastic mean-free path (IMFP). Since the areas of Voigt function lines are not proportional to the product of peak height and Gaussian width, it will be necessary to numerically integrate the area to set the peak amplitude. In the buckled dimer reconstruction of the Ge(100) surface, each layer contains an equal number of atoms. Those that are in the surface layer exist in two states in equal number. In Si(001), there are features that divide the atoms in both the first and second subsurface layers into two classes (Figure 1). However, the atoms in the second subsurface layer have been considered to be identical in Ge(001), which is in contrast to the case in Si(001). In other words, only the first two top surface layers of Ge(001) are regarded as behaving differently from the bulk.

The algorithm that has been successfully set up to analyze the complicated line shape of the Si 2p core-level spectrum will be mainly used here to analyze the Ge 3d core-level spectra. The details can be found elsewhere [21,22]. In brief, the assumption of layerwise attenuation through the parameters of the escape depth (λ) and the layer spacing (*d*) correlated by *x* = exp(−*d*/λ) gives the fractional areal strengths of the first and second layers to be 1 − *x* and *x* × (1 − *x*), respectively. Given that the λ value is approximately 4 Å at a photon energy of 80 eV, the relative areal intensities, in the fit of the first layer (S1), the second layer (S2), and the bulk (B) in the normal-emission spectrum are expected to be 28%, 24%, and 48%, respectively. Due to the layerwise attenuation models only estimating the maximum fractional intensity of the second layer as 0.24, an unconstrained model function that gives a larger fractional intensity for the second layer can be immediately discarded.

## 3. Results and Discussion

### 3.1. Clean epi Ge(001)-2 × 1 Surface

The surface atoms naturally exhibit electronic structures that are distinct from the bulk atoms. Core-level photoemission is highly sensitive to the surface electronic structure, which is manifested by a SCLS from the bulk. If the surface line lies in a lower binding energy than the bulk, the shift is conventionally regarded as a negative shift, if higher, then the shift is said to be positive. In the specific Ge studies, efforts have been made to interpret the shallow Ge 3d core-level spectrum [23,43,44,45,46,47]. The lack of consensus in the SCLSs of the dimerized surface atoms in the established reports is rather unusual, despite the similar line shapes in these room-temperature data and fewer contributed line components in the Ge 3d state than those in the Si 2p state [22,44]. Some authors have suggested that the dimers are unbuckled, thereby giving rise to only one S1 component in the spectra [43]. The proposals for a dimerized surface differ; the shifted signs of the down atoms have been reported as positive [43,44,45] and as no shift at all [46,47]. It is not only the surface component(s) that have been reported differently by various research groups; there is also disagreement between groups regarding the line position of the second surface layer, which has been reported as having both a positive [23] or negative shift [43,44,45,46,47]. As follows, we show a resolution by approaching the line shape of an *epi* Ge(001)-2 × 1 surface assisted with the physical IMFP effect. Note that the existing reports that are cited above dealt with chemically treated surfaces.

Figure 2a displays SRPES-acquired *epi* Ge 3d core-level spectra in normal and 65° off-normal emissions at room temperature. It is worth noting that the data presented in Figure 2 were collected after the samples had been in the portable UHV module for 48 h. The absence of an oxidation state suggests that the *epi* Ge(001) surface is rather stable in vacuum. If a similar arrangement is applied to Si(001), then the surface dangling bonds will certainly be oxidized.

As shown in Figure 2a, a component lies clearly on the low-energy shoulder of the bulk component. Moreover, a change in line shape with emission angles is also observed, which is in contrast to the literature, where is it reported that little change in the spectral line shape occurs at different energies and emission angles [23,44,45,46]. Namely, the valley region increases in strength with increasing emission angles. Clearly, the *epi* Ge(001) surface shows a distinct surface electronic structure by SRPES that cannot be overlooked upon dielectric deposition. Note that the acquired XPS spectrum becomes broadened due to poor energy resolution, and the IMFP effect renders the contribution of surface emission rather small [42].

A fit is necessary to extract the embedded components of the *epi* Ge 3d line shape. A preliminary analysis that uses a proper model function suggests the need for four Voigt function-like components, which reside on a parameterized background and separate energy-loss tail. Four parameters represent the background: a constant, a slope, and a two-parameter power-law. This background function successfully represents the attenuated electron from a shallow Ge 3d level, where the loss tail is due to the electron-hole pair excitations in the spectral energy range. The presence of such a tail does not affect the primary line structure, because the onset of electron-hole pair excitations commences at 1.2 eV away from the bulk line. Note that the background function must be incorporated in a fit. Subtraction of the background before a fit should be avoided as it ignores the electron-hole pair excitations.

Figure 2b,c for θ_e_ = 65° and 0°, respectively, plot a representative fit to the room-temperature data, respectively. In this fit, the spin-orbit splitting is 0.593 and the spin-orbit branching ratio varies greatly from 0.514 in normal emission to 0.626 in θ_e_ = 65°. The lifetime width is 0.172 eV. The low branching ratio has been explained as being due to the photon energy bearing near the photoemission threshold [48,49]. The binding energy of the bulk Ge 3d state is resolved at 29.25 eV. The SCLSs of the S1u, S1d, and S2 components are −0.462, −0.159, and +0.165 eV in the normal emissions, respectively. The S2 component solely originates from the first subsurface layer [21,23]. The resultant intensities of the three surface components follow the expectation from the IMFP effect.

Decades of researches examining Si(001)-2 × 1 and Ge(001)-2 × 1 came to the consensus that the final-state theory explains the Si 2p and Ge 3d core-level spectra better than the initial-state theory. The final-state picture includes a crystal ensemble with a created core hole and a photo-excited electron. Pehlke and Scheffler [50] calculated the final-state values for the S1u and S1d components to be −0.67 and −0.39 eV away from the bulk, respectively. For one’s reference, the initial-state values for the S1u and S1d components were −0.50 and +0.27 eV, respectively. A large screening shift was found for the 3d core electrons that were emitted from the down-dimer atoms. This was hypothesized to be the unoccupied dangling-bond state coming from the down-dimer atoms that were pulled down due to the influence of the core hole. The down-shifted dangling-bond state becomes populated by electrons from the Fermi level, thereby giving rise to effective screening. The lower value that was found in the present work suggests imperfect screening in the physical process [50]. Note that the phenomenon is less apparent in the 3d core electrons from the up-dimer atoms, because the corresponding dangling-bond state is already occupied. The final-state model predicts that the SCLS of the S2 component has a negative line position [50,51], which disagrees with the results that are presented in Figure 2a.

### 3.2. Adsorption of Molecular Oxygen on epi Ge(001)-2 × 1 Surface

Figure 3a shows the Ge 3d core-level spectra with various dosages of O_2_ on the *epi* Ge(001)-2 × 1 surface at room temperature, being taken with 80-eV photon energy in normal emission. The inset plots the O 1s core-level spectra with selective O_2_ dosages, where two O 1s states, O(I) and O(II), are resolved 1 eV apart. As displayed in Figure 3a, the adsorption of O_2_ alters (not drastically) the line shape of the Ge 3d state, and the development of a germanium-oxide state on the high binding energy side of the bulk state is not as significant as that of SiO_x_ in Si(001)-2 × 1. The latter shows equally spaced peaks that gradually increase in intensity with increasing O_2_ dosage [24]. The molecular oxygen immediately affects the S1u component in Figure 3, suggesting that the oxygen makes direct contact with the up-dimer atoms. As can be seen in Figure 3a, the emission from the S1u state becomes broadened but it remains observable at great coverages. Attempting to have O_2_ covering all of the surface-dimerized atoms is unlikely, even 1000 L of O_2_ exposure is of little help here [52]. In other words, the O_2_/Ge(001)-2 × 1 interface is unlikely to result in high-oxidation states. In comparison, just 10 L of O_2_ on Si(001)-2 × 1 have already developed four oxidation states with the highest one at the surface region and the smallest one on the bottom of the oxide layer [24]. Note that the surface Si-Si dimers remain detectable at 10 L of O_2_ dosage, which suggests the local distribution of the SiO_x_.

The affected Ge atoms would either appear as an induced component in the spectra or be removed from the surface. The induced Ge-O component would show increased binding energy below the bulk component, which is embedded in the spectral line envelope. Other than the noticeable feature lying an eV from the bulk line, another component that sits in the valley region is realized if the spectra of Figure 3a are normalized to the bulk line. Hence, six components are needed to represent the Ge 3d core-level spectra in the model function, four for the O_2_-free, and two for the O_2_-affected atoms. The number of O_2_-induced components is consistent with two O 1s states, as shown in the inset of Figure 3a.

Further analysis of the line shape with a fit would allow us to determine whether the contacted oxygen is in an atomic or molecular form. Figure 3b,c display a fit to the 80 L and 2 L curves of Figure 3a, respectively. The induced GeO(I) and GeO(II) components are, respectively, resolved at approximately +0.355 and +1.333 eV, which gradually increase in intensity with increasing O_2_ dosages. One might question the necessity of the GeO(I) component in Figure 3c, because its intensity is rather small. If the GeO(I) component was excluded in the model function, the S1d component would end up with an unphysical enhancement in intensity (~20%). The appearance of the GeO(I) component in the embryonic stage of O_2_ adsorption suggests the immediate attachment of O_2_ onto the up-atom dimers upon arriving at the Ge(001) surface. That is to say that no mediate stage exists in the O_2_ adsorption on Ge(001) [53]. In a fit, the S2, as well as the S1u component, shows a gradual decrease in intensity with O_2_ dosages.

In a separate experiment, the deposition of a high-κ dielectric oxide onto the present 300-L O_2_/Ge(001) surface has reduced the GeO(II) component (data not shown), which has a behavior that is similar to that of GaAs(001). That is to say, the GeO(II) is not an interfacial O/Ge component. The phenomenon indicates that the Ge atoms in GeO(II) are those that originated at the S1u up-dimer atoms now being set free to become the GeO(II) film. Once an S1u atom is freed, the other dissociated O atom immediately fills the vacancy and then bonds with the S2 atom underneath, thereby giving rise to the GeO(I) component. As shown in Figure 3b,c, the S1d component remains virtually in the same areal intensity as in the clean surface, which suggests that it does not follow the reacted pathway of the S1u component. Figure 4a presents the schematic drawings that summarize the interaction of O_2_ with an up-dimer atom. The drawing is self-explanatory; the incoming molecular oxygen would disrupt the Ge-Ge dimer, so that one of the oxygen atoms would remove the up-dimer atom and the other immediately bonds with the nearby underneath subsurface Ge atom.

The preferential inclination of the O_2_ molecules onto the up-dimer Ge atoms is certainly not found in the silicon counterpart and the previous reports of the chemically treated Ge surfaces [33,34,36,37]. On the one hand, the non-dissociative model should be forsaken, because it would result in only one GeO and one O 1s component. The O_2_-dissociative model, on the other hand, predicted a non-destructive effect on the Ge-Ge dimers, with one oxygen atom sitting at the top bridge site and the other the backbond of the surface dimer. This assignment has the fundamental difficulty that the bridged O would revert the buckled dimer to a symmetric configuration, and a change in the work function should consequently be observed due to the change of the dipole within the dimerized atoms. Our separate measurements of the valence-band spectra and the cutoff region show gradual increases of the O 2s state and an unshifted zero kinetic-energy point with various O_2_ exposures, respectively. The latter indicates that the work function of the O_2_/Ge interface stays at the same value as that of the clean surface.

### 3.3. Adsorption of Atomic Oxygen on epi Ge(001)-2 × 1 Surface

Figure 5 displays a series of atomic-O covered *epi* Ge 3d core-level spectra that were taken with 80 eV photons at normal emission at room temperature. The dense plots in the bottom part of Figure 5 are the result of finely incremental O dosages below 1.86 L, and the upper part of the figure shows the curves over a dosage of 50 L. As can be seen in Figure 5, a gradual decrease in the surface intensity of the S1u state with increasing atomic O coverages suggests an immediate affiliation of the oxygen atoms with the surface dimers. As can be seen in Figure 5, the 400-L spectrum is not significantly different from the 1.86-L spectrum, suggesting that the growth of atomic O on *epi* Ge(001)-2 × 1 is a self-limiting process. Furthermore, the Ge 3d spectra display a peak shift of 40 meV towards a lower binding energy upon early appearance of oxygen atoms, which is later compensated for when the surface dimers have adsorbed the maximal amount of atomic O.

Figure 6 displays a fit for the representative curves reflected in Figure 5. Upon the early appearance of atomic O on the *epi* Ge(001)-2 × 1 surface, the line shape shows no great difference from that of the clean surface. Hence, we employ the model function of the clean surface to analyze the O-affected line spectra with great success. In contrast to the case of O_2_ on Ge(001)-2 × 1 above, the O-bonded Ge atoms do not occur in an energy position that is lower than the bulk atoms, but they appear on the two dimer-related components. Note that the increased binding energy under expectation should be referred to the S1u component, not the bulk component. As can be seen in Figure 6, the (S1u’+S1d) component gradually increases in strength at the expense of the S1u component. Moreover, the S2 component remains virtually unaltered in intensity, even after the 400 L exposure to atomic oxygen (data not shown). The O atoms confine the reaction to the topmost surface layer and even restrict themselves to the top of the up-dimer atoms (see later). The oxidation path of atomic O on Ge(001)-2 × 1 certainly develops on its own, and it does not behave similarly to that of O_2_ on either Si(001)-2 × 1 or Ge(001)-2 × 1. Oxygen is readily available to accept a full charge from Si to allow the interface to form a SiO_x_ film with x = 1 to 4 [24]. However, only a partial charge of the Ge surface atoms is needed to transport to the adsorbed oxygen atoms.

As the (S1u’+ S1d) symbol suggests, it consists of two states, the S1d atoms and the other from the O-bonded S1u atoms (S1u’), which makes the component 50% wider (e.g., 0.48 L) than the S1d component on the clean surface. The O-bonded S1u component appears to be fairly close to the S1d component in terms of energy position, thereby doubling the line intensity in the region. Attempts to split the S’ component into two components yielded too small of an intensity in the induced S1u component. Direct evidence to support this speculation is derived from the information that is provided by the surface dipole. As is known, the buckled dimer results in a charge transfer from the down-dimer atom to the up-dimer atom, thereby giving rise to a dipole moment that was oriented with the positive end inwards in the asymmetric dimer. This will lead to an increase in the ionization potential (IP) of the buckled surface when compared with the unbuckled Ge(001)-2 × 1 surface. The change in IP upon O adsorption suggests an induced dipole. Due to the atomic oxygen accepting a charge from the up-dimer atom, the positive end of the O-Ge dipole moment orients the Ge atom. As a result, the IP becomes large in magnitude when the O atom resides on the up-dimer atom. The IP in a semiconductor is determined through the measurement of the spectral width (W) at a given photon energy (hν). The former W is the energy separation of the valence band maximum (VBM) and the cutoff of the photo-ejected electrons. The IP value is directly determined without any assumption by subtracting hν from W [54,55].

Figure 7 plots the spectral development of the cutoff and valence band regions. As shown in Figure 7, the onset of the cutoff gradually moves towards the lower kinetic energies with increasing atomic O dosages. The extrapolation of the rising slope with the constant background line pins down the cutoff position, and the IP development of each dosage with the nearly fixed valence band maximum (VBM), is plotted in the inset of Figure 7. The IP value decreases from 5.31 eV in the clean surface to 5.02 eV after 0.78 L of atomic O dosage. The result of the decreased IP is contrary to the expectation that the O atom is above the Ge atom. Note that the final IP occurs when the atomic O bonds with the up-dimer atoms fully. Although the binding energies of components S1u’ and S1d are close to each other, the O-bonded dimer is still manifested in the buckled orientation. The placement of the atomic O at the backbond site is in accordance with the upward dipole direction that is found in the present study, as well as the theoretical calculation [56]. In the latter, it has been claimed that the highest energy-occupied surface states were exclusively backbond states [56]. Figure 4b shows the schematic drawing of atomic O/Ge(001)-2 × 1 bonding configuration. As shown in Figure 6e,f, the GeO(I) and GeO(II) components begin to appear at the Ge(001) surface, meaning that the O_2_ has come into play on adsorption. This is because the present setup for breaking the molecular oxygen into its atomic form is not 100% efficient, according to the mass spectrum from a residual gas analyzer. However, the small O_2_ residual does not affect the IP of the system, as mentioned above.

As can be seen in Figure 6, both the S1d and S2 components maintain the original energy positions with the unchanged SCLS signs in the atomic O-covered spectra. This suggests that the topmost dimers layer serves as a charge-enriched layer, while the subsurface layer serves as a charge-deficient layer. This indicates that a charge redistribution occurs between the first two surface layers. If this is indeed the case, then the Ge 3d core-level spectra faithfully show that the sign of the shift is negative in the first surface layer and positive in the second surface layer.

## 4. Conclusions

An early stage of oxidation of an *epi* Ge(001)-2 × 1 surface by atomic as well as molecular oxygen is presented while using high-resolution synchrotron radiation photoemission as a probe at room temperature. The pristine reconstructed Ge(001)-2 × 1 surface was first presented by setting up a proper model function to analyze the Ge 3d core-level spectrum, which is necessary for the later observation of oxidation development. In fact, only the first two top surface layers are considered to behave distinctly from the bulk. However, the topmost surface is buckled with one atom moving upward and the other downward. A charge transfer indeed occurs in between with the up-dimer atom being in a charge-enriched state, and the down-dimer atom a charge-deficient state. The charge environment plays a significant role for the oxygen atoms to preferentially react with the up-dimer atoms. For O_2_, it is immediately dissociated without a mediated stage, and it simultaneously causes the up-dimer atom to exit the dimerized state. A dissociated oxygen bonds with the freed Ge atom, and the other inclines to be positioned at the site of the freed up-dimer atom, and bonds with the underneath Ge atom in the subsurface layer. The down-dimer atoms and those in the subsurface layer are inert to O_2_. For atomic O, it selectively sits at the position between the up-dimer atom and the atom in the subsurface layer without causing any bond to be broken. Similar to the O_2_ case, the down-dimer atoms are resistant to the effect of atomic oxygen. The reconstructed 2 × 1 configuration remains intact upon atomic O adsorption. Unlike the O/Si case, the adsorbed oxygen atoms accept a partial charge from the contacted Ge atoms. The full coverage of the up-dimers atoms with oxygen renders the surface layer into a uniform electronic state with excess charge. The second surface layer stays in the electronic state with deficient charge. The experimental results presented here might explain some of the reliability problems that are associated with the Ge MOS devices.

## Figures and Tables

**Figure 1 nanomaterials-09-00554-f001:**
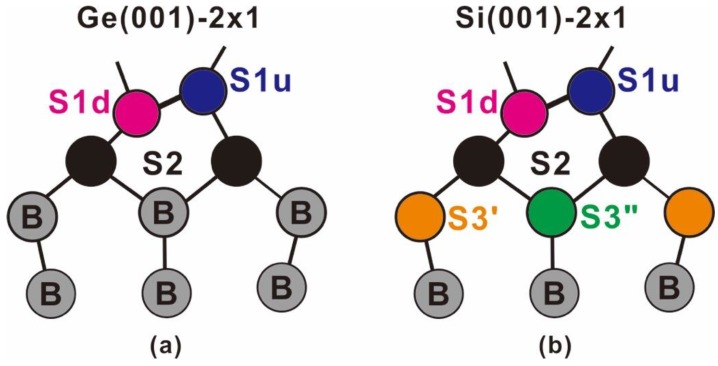
Schematic side-view drawings of (**a**) buckled Ge(001)-2 × 1 and (**b**) buckled Si(001)-2 × 1 surface. Symbols S1u, S1d, S2, and S3 stand for the up-dimer atom, down-dimer atom, atoms in the second surface layer, and atoms in the third surface layer, respectively. In Ge(001)-2 × 1, the S3 atoms show one electronic environment, but in Si(001)-2 × 1, they are differentiated by S3’ and S3’’ electronic environments.

**Figure 2 nanomaterials-09-00554-f002:**
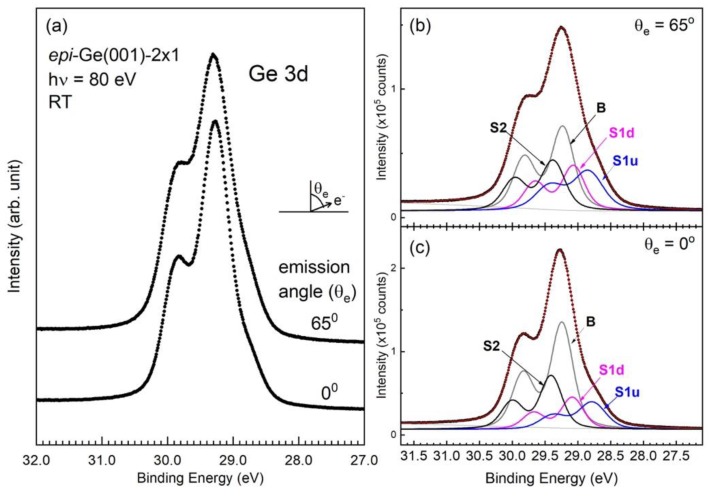
(**a**) Photoemission data from as-grown *epi* Ge(001)-2 × 1 taken in normal (θ_e_ = 0°) and off-normal (θ_e_ = 65°) emission at photon energies (hν) = 80 eV at room temperature. (**b**) A simultaneous fit to the 65°-emission spectrum. (**c**) A simultaneous fit to the normal-emission spectrum.

**Figure 3 nanomaterials-09-00554-f003:**
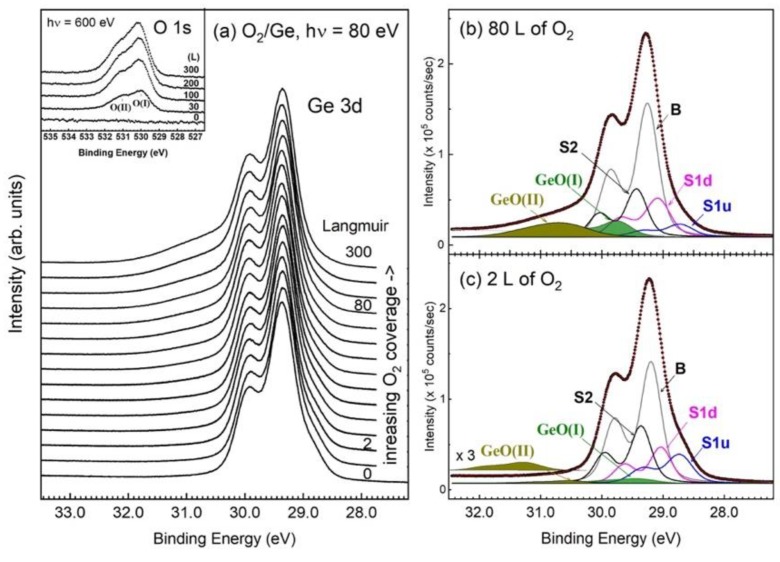
(**a**) The Ge 3d core-level spectra with various dosages of O_2_ on an *epi* Ge(001)-2 × 1 surface at room temperature; (**b**) a fit to the Ge 3d core-level spectrum with 80 L of O_2_ on Ge(001)-2 × 1; and, (**c**) a fit to the Ge 3d core-level spectrum with 2L of O_2_ on Ge(001)-2 × 1. (Panels (**a**) and (**c**) reprinted with permission from [42], Copyright The Japan Society of Applied Physics, 2018.)

**Figure 4 nanomaterials-09-00554-f004:**
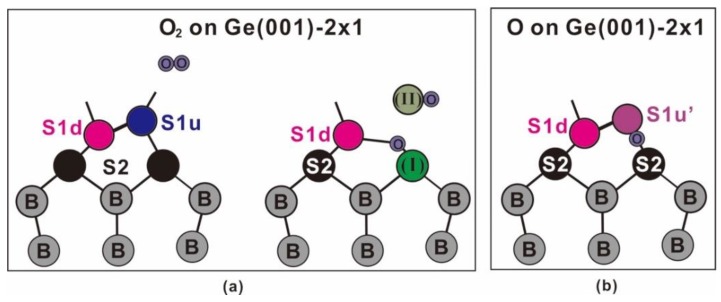
Schematic drawings of (**a**) O_2_ onto Ge(001)-2 × 1, and (**b**) atomic O on Ge(001)-2 × 1 bonding configuration.

**Figure 5 nanomaterials-09-00554-f005:**
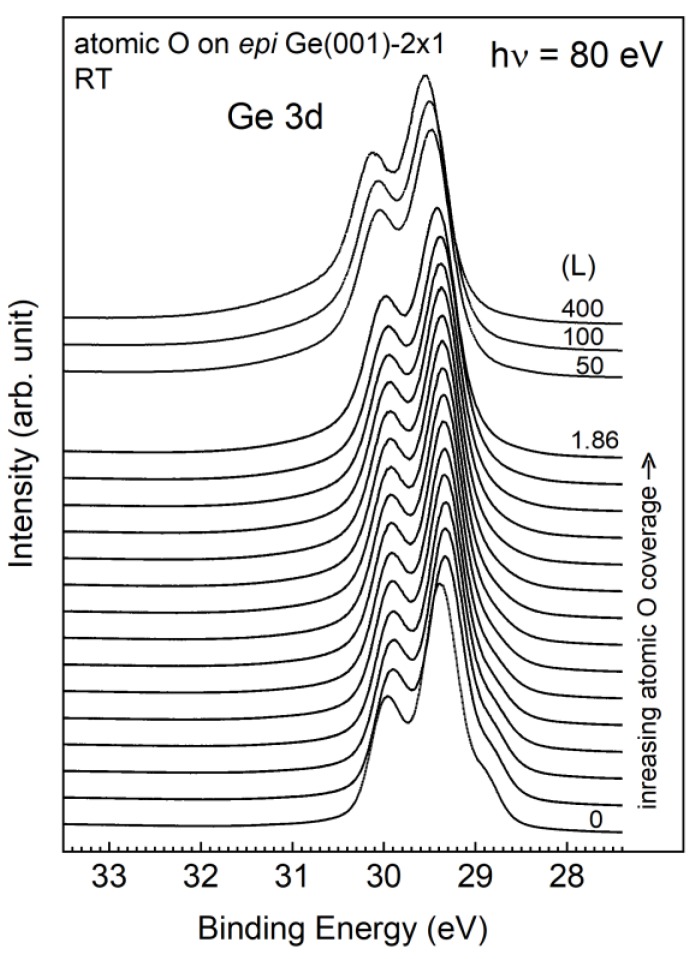
Development of the Ge 3d core-level spectra with various dosages of atomic O on an *epi* Ge(001)-2 × 1 surface at room temperature.

**Figure 6 nanomaterials-09-00554-f006:**
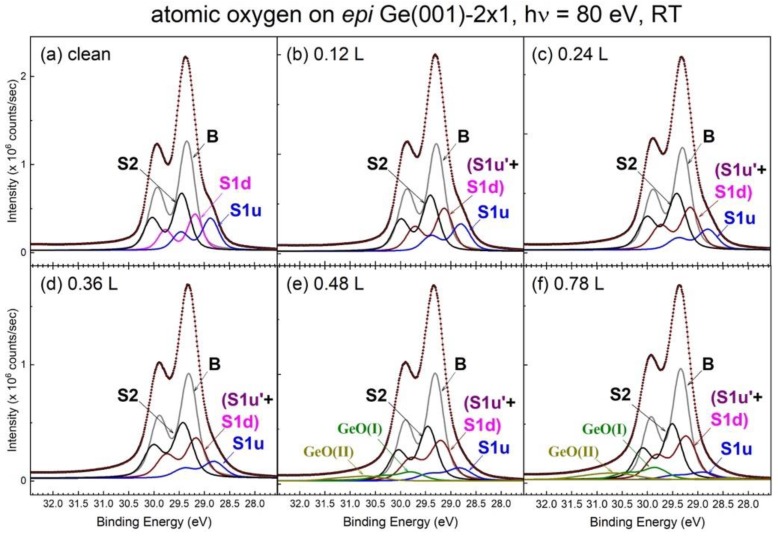
A fit to the representative Ge 3d core core-level spectra in Figure 5 emphasizing the initially low atomic-O coverages.

**Figure 7 nanomaterials-09-00554-f007:**
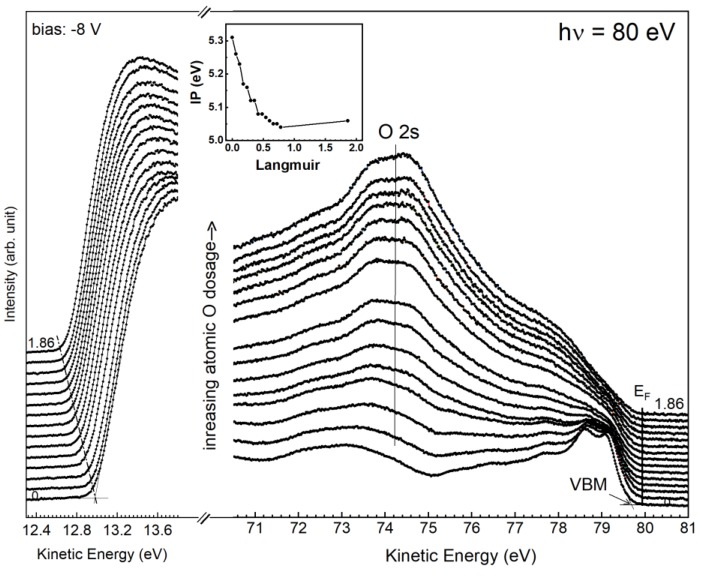
The acquired valence-band and cutoff spectra taken with 80 eV photon energy with various atomic-O coverages. The inset shows the change of the ionization potential (IP).

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
