# Peer review of "Microscopic Views of Atomic and Molecular Oxygen Bonding with epi Ge(001)-2 × 1 Studied by High-Resolution Synchrotron Radiation Photoemission"

_nanomaterials, 2019, doi:10.3390/nano9040554_

Reviewer 1 Report

The authors studied the initial oxidation process of Ge(001) by tracking the surface core-level shift and the intensity change in the Ge 3d photoemission line.  The experiment was done carefully and the conclusion is well supported by the spectral analysis. A few points, see below, can be clarified to improve this manuscript.

Please mention the main cause of the difference in spectra taken at two different emission angles? Does IMFP explain it reasonably? Indicating the IMFP values at these emission angles will be helpful. 

The authors assumed that there was no photoelectron diffraction effect. This can be evidenced by analyzing a spectrum taken at a different excitation photon energy. If IMFP explains the intensity variation between components there, it means there is a negligible diffraction effect. 

Author Response

The IMFP effect could be realized its significance in the photoelectron spectra taken with various photon energies in normal emission. See our precious report in Journal of Electron Spectroscopy and Related Phenomena Vol. 107, p163 (2000) in details.We did that for epiGe(001)-2×1,but only present the 80-eV one here because it among others is the most surface-sensitive scan. The additional angled off-normal emission spectra serves for two purposes. One is to justify the employed model function in a fit, and the other to pinpoint the topmost surface-related peaks which would be much enhanced in intensity. The IMFP for the normal-emission 80-eV spectrum in the present epiGe(001)-2×1is as short as 4 angstroms. Mentioning the IMFP for the off-normal emission spectrum becomes redundant. Besides, they are merely related by a cosine function.

The photoelectron diffraction effect would cause an intensity variation in a periodic basis. It does not happen to the present Ge(001)-2×1 as not in the case of Si(001)-2×1 under the VUV energy range. In fact, the photoelectron diffraction effect is unlikely to happen in Ge(001)-2×1 and Si(001)-2×1 because the surface buckled dimers would kill the coherent wave from the underneath layers. 

English has been further proofread by a manuscript editing company.

Reviewer 2 Report

This paper reports a detailed study of tracking the oxidation process of a well defined Ge(001) 2X1 surface using synchrotron radiation photoemission. While the technique in tracking core level binding energy shift has been around, to do a good job requires careful control of the preparation of the surface under study, and a good match of photon energy for surface sensitivity (cross-section and escape depth of the photon electrons as well as the experimental geometry and polarization when applicable). This paper meets these requirements

The selection of 80 eV for Ge 3d core level makes the technique surface sensitive (photoelectron will have KE of about 45 eV, at the escape depth minimum, can see only serval layers,); 65o emission makes it even more surface sensitive. Changing depth sensitivity could also have been done with changing photon energy. The analysis is done properly, and the assignments of the peaks are reasonable. Given an energy resolution of 60 meV, I would have expected to see sharper peaks; perhaps there are some surface defects that broadens the peaks. I am still puzzled by the +ve 3d shift of the S2 assignment; I would have expected it to be -ve relative to the bulk. Also, for surface like this, LEED would have been better than RHEED.

The exposure of atomic oxygen shows a clear effect on S1u as the authors explained, the assignment of the O2 exposure is less obvious, while it is consistent with the fitting, there could be other possibilities. The authors should perhaps describe a bit more on how the fitting is done (Fig.6). Obviously, fitting 4 to 5 doublets under one doublet with a shoulder is not trivial and needs good modeling.

The O 2S in Fig. 7 appears to be a doublet, would this be due to the exchange splitting of O2 adsorption, or the presence of para-magnetism, this should be clarified.

In summary, the authors have done a careful job in surface preparation and photoemission characterization, the assignments are generally consistent with the observation. The less satisfactory assignment is the S2 3d shift, which appears to be +ve relative to the bulk and is also insensitive to surface oxidation. Some clarification should be provided in the revision.

This paper can be accepted after minor revision in which the authors have taken into account the above comments at their discretion.

 Author Response

The response letter includes figures to properly answer the reviewer's comments.  Please open the attached file to read the responses. 

Reviewer 3 Report

This manuscript contains some high quality XPS spectra from atomic and molecular oxygen on Ge (001). If one believes the fitting, which builds on previous work and hence has some validity, some interesting conclusions might be possible from this work. However, the paper is so difficult to read, with strange language expressions, that is too difficult to understand what was done, the discussion and if the conclusions make any sense. The manuscript needs a radical rewrite to enable it to be reviewable.

Author Response

We were presenting the core-level spectra taken by synchrotron radiation photoemission in the VUV range, where the kinetic energy of the excited photoelectrons lies in an IMFP value about 4 angstroms. The VUV spectrum is much higher surface sensitive than the XPS spectrum, which is hard to provide clear information of surface and interfacial electronic structure. The surface-related peak(s) often shows a small shift from the bulk, which hides itself under a broad envelop. In the past three decades, the SRPES society has commonly used a fitting algorithm to extract the hidden surface components with great success. We have been using it to study the metals as W, single-crystal semiconductors as Si, GaAs, Ge, and the interfaces thereof, and obtained noted records in the past nearly 30years. We must stress that the fit is not merely a mathematical act, but involves much of the physical meanings with the theoretical supports as it should be. By taking advantage of the IMFP effect to determine the layered intensity, for example, we have resolved the long-puzzled Si 2p cores of Si(001)-2×1 and Ge 3d cores of Ge(001)-2×1, not to mention other pioneered works as GaAs(111)A-2×2 , GaAs(001)-4×6, GaAs(001)-2×4, and InGaAs(001)-4×2 surfaces. In particular, the final-state effect dominates the Si 2p as well as Ge 3d core-level spectra, and the Ga-rich GaAs(001)-4×6 surface is terminated with the As atoms, not the Ga atoms. 

The reviewer comments “that it too difficult to understand what was done, the discussion and if the conclusions make any sense.”  It should be noted that the basic researches of clean Ge(001) and oxidation of Ge(001) are rather limited, and we have included the relevant references in the “Introduction” of the manuscript. 

The reviewer also comments that language expressions are strange. This point puzzles us because we have been using the language expressions in the SRPES circle for the past nearly 30years. Nevertheless, the English is improved with the paper been proofread by a global manuscript editing company twice. Hope that the reviewer might read well the resubmitted version. 

Round  2

Reviewer 3 Report

As described in the previous report the manuscript was difficult to read due to a poor level of English in many parts of the manuscript. The new version is much better and certainly reviewable. The results are interesting, the data of high quality and the explanation plausible. With a little more work on the wording, which will help the reader to follow the paper, the present version can be accepted. Nevertheless, below are marked some rewording suggestions and a few cases where the meaning is still unclear to the reviewer. The manuscript will be much better if these small changes are made.

1.       Use “embryonic” rather than “embryo” throughout the document.

2.       Line 50: …..communities, especially …. Circle,… [add commas]

3.       Line 52: understanding - singular.

4.       Line 55: intensive research - singular

5.       Lines 84-85: ….pressure of O2 is immune to the… ?? Meaning unclear?? The O2 is unreactive with the (In)GaAs substrates?  Suggestion.

6.       Lines 100-101: Note that the present work does not deny the established XPS works because, .. Meaning? The present work is not inconsistent with the previous XPS results?

7.       Line 109: remove “generously”

8.       Line 149: analyze the complicated - remove much-

9.       Line 150: used here to analyze the Ge 3d…

10.   Lines 171-173: It is not only the surface component(s) that have been reported differently by various research groups; there is also a disagreement between groups about the line position of the second surface layer, which has been reported as having both a positive [23] or negative shift [43-47].

11.   Line 198: "attenuated" instead of "degraded"?

12.   Line 203: the electron-hole pairs remove "physical".

13.   Line 207: The low branching ratio has been explained as being due to the photon energy bearing near to the photoemission threshold [48,49].

14.   Line 210: The S2 component originates solely from the first subsurface layer [21,23].

15.   Line 219: ….dangling-bond state coming from the down-dimer atoms pulled down due to the influence of the core hole.

16.   Line 221: "lower value" instead of "undervalue"?

17.   Lines 224-225: The final-state model predicts that the SCLS of the S2 component has a negative line position [50,51], which disagrees with the results presented in Figure 2(a). ?

18.   Line 238: interface is unlikely to result in high-oxidation states.

19.   Line 264: embryonic

20.   Line 265: That is to say no intermediate stage exists. ???suggestion.

21.   Line 269: onto the present 300-L O2/Ge(001) 268 surface has reduced the GeO(II) component? Suggestion.

22.   Lines 277-278: The drawing is self-explanatory; the incoming molecular oxygen would disrupt the Ge-Ge dimer so that one of the oxygen atoms would remove the up-dimer atom ….

23.   Line 284: ….effect on the….

24.   Lines 291-292: Moreover, it is known that the distribution of the Ge charge state in an oxide film would run less in the bulk than in the surface [24,53]. What does this mean? Rewording needed.

25.   Line 317: Note that the increased binding energy under expectation should be in reference to the S1u component. What does this mean? Rewording needed.

26.   Line 336: As is known, the buckled dimer results in a charge transfer from the….Suggested change.

27.   Line 362: the mass spectrum from a residual gas analyzer.

28.   Line 393: suggest “uniform” rather than “uniformly”.

Author Response

We thank the reviewer to edit the paper to make it a better one. Below we answer a few comments. 

Points:

5.       Lines 84-85: ….pressure of O2 is immune to the… ?? Meaning unclear?? The O2 is unreactive with the (In)GaAs substrates?  Suggestion.

Response:

As suggested. Change to “unreactive with the (In)GaAs substrates”. Exposure of O2to (In)GaAs under UHV does not show any O-induced peak.

6.       Lines 100-101: Note that the present work does not deny the established XPS works because, .. Meaning? The present work is not inconsistent with the previous XPS results?
Response: The established XPS work (Ref. 31) showed two O-induced peaks. The poor resolution, however, is unable to observe the atom-to-atom interaction at the interface.

20.   Line 265: That is to say no intermediate stage exists. ???suggestion.

Response: We add a reference numbered 53, which suggested a mediated-trapping adsorption of O on Ge.

24.   Lines 291-292: Moreover, it is known that the distribution of the Ge charge state in an oxide film would run less in the bulk than in the surface [24,53]. What does this mean? Rewording needed.

Response: We were referring to our precious work of oxidation of Si(001)-2x1, which showed gradient distribution of SiOx (x = 1 to 4) from the surface down. The sentence is deleted in order not to make confusion. 

25.   Line 317: Note that the increased binding energy under expectation should be in reference to the S1u component. What does this mean? Rewording needed.
Response: Rewording as “Note that the increased binding energy under expectation should bereferred tothe S1u component, not the bulk component.”
